# A Novel Method of Hyperspectral Data Classification Based on Transfer Learning and Deep Belief Network

**Ke Li [1], Mingju Wang [1], Yixin Liu [1], Nan Yu [1] and Wei Lan [2],***

[1] Fundamental Science on Ergonomics and Environment Control Laboratory, School of Aeronautics Science and Engineering, Beihang University, Beijing 100191, China; like@buaa.edu.cn (K.L.); wangmingju@buaa.edu.cn (M.W.); yixinliu@buaa.edu.cn (Y.L.); yunan@buaa.edu.cn (N.Y.)

[2] School of Economics and Management, Beihang University & Acadamic Affair Office, Beijing 100191, China

\* Correspondence: lanwei@buaa.edu.cn; Tel.: +86-010-8231-6654

**Abstract:** The classification of hyperspectral data using deep learning methods can obtain better results than the previous shallow classifiers, but deep learning algorithms have some limitations. These algorithms require a large amount of data to train the network, while also needing a certain amount of labeled data to fine-tune the network. In this paper, we propose a new hyperspectral data processing method based on transfer learning and the deep learning method. First, we use a hyperspectral data set that is similar to the target data set to pre-train the deep learning network. Then, we use the transfer learning method to find the common features of the source domain data and target domain data. Second, we propose a model structure that combines the deep transfer learning model to utilize a combination of spatial information and spectral information. Using transfer learning, we can obtain the spectral features. Then, we obtain several principal components of the target data. These will be regarded as the spatial features of the target domain data, and we use the joint features for the classifier. The data are obtained from a hyperspectral public database. Using the same amount of data, our method based on transfer learning and deep belief network obtains better classification accuracy in a shorter amount of time.

**Keywords:** hyperspectral data; deep learning; transfer learning; DBN

## 1. Introduction

Hyperspectral data are three-dimensional data that are composed of data obtained from hundreds of spectral channels [1]. With the development of hyperspectral sensors, the hyperspectral data can have a wider range of wavelengths and more bands. Hyperspectral data contain spectral information in different bands, which provides hyperspectral data with outstanding advantages in object recognition [2]. Therefore, hyperspectral data are widely used in vegetation ecology, atmospheric science, geology, mineral resources, marine research, military detection and recognition, camouflage, and other areas [3–7].

As the performance of hyperspectral sensor hardware improves, the acquired hyperspectral data are improved in spectral resolution, but also greatly improved in spatial resolution, while down sampling can cause a rapid loss in spatial detail [8]. This improvement is also accompanied by the emergence of certain problems. First, as the band of hyperspectral data increases, more spectral information can be provided, and the accuracy of the object recognition increases. However, when the spectral band is increased to a certain extent, the accuracy of recognition will instead be influenced. In other words, a "dimensionality disaster" [9] occurs due to the redundant information caused by too much band information. On the other hand, with the improvement in spatial resolution, the number of pixels collected by the hyperspectral data gradually increases. In the process of the classification

and identification of hyperspectral data, calibration of the corresponding artificial labels will result in a large amount of manpower and a loss of financial resources [10].

There are many research methods to address dimensionality disasters and other issues at the early stages. For feature selection and feature extraction in the hyperspectral data spectral dimension, in the earliest research on the feature extraction of hyperspectral data, the main research methods, such as principal component analysis (PCA) [11], independent component analysis (ICA) [12], and linear discriminant analysis [13], all use linear models to extract features. However, hyperspectral data are non-linear. Therefore, in 2006, Hinton improved deep neural networks and solved the problem of gradient diffusion and the local extreme value of deep neural networks by means of layer-by-layer greedy training [14].

Afterwards, deep learning networks were gradually applied in a series of feature extraction and classification approaches to hyperspectral data. Among these approaches are the deep belief network (DBN) [15], stacked autoencoding (SAE) [16], denoising autoencoding (DAE) [17], and conventional neural networks (CNNs) [18,19]. The applications of these deep learning networks (with the adjustment of the number of network layers and the number of neuron nodes) can yield a series of features of hyperspectral data from low levels to abstraction [20]. While the deep learning network structure can obtain very effective features, a deep learning network conducts training for a long time; thus, it takes a large amount of manpower and financial resources to sample the data calibration [21]. However, if unsupervised learning is used for the network training, then there are disadvantages, such as low classification accuracy [22].

This paper proposes a novel deep learning network model that uses the transfer learning approach. A part of the model that has been trained in the hyperspectral data in one field is proposed. For a new field of hyperspectral data, if there is some connection or similarity between the hyperspectral data of these two fields, then we do not need as much data to study the hyperspectral data set of the new field. We would only need a small amount of data to complete the network training. Therefore, based on the idea of transfer learning, we can realize that the new model requires a very small amount of sample data, and does not need to repeatedly train all of the deep learning networks. This would achieve the goal of saving manpower, materials, and financial resources through the use of a smaller number of samples and the utilization of the transfer learning model. This approach also reduces the network training time.

A novel model of the classification method that we proposed is mainly divided into the following steps. The first step is to construct a deep learning network model using source-domain hyperspectral data with a large number of tags and perform training to extract the abstract features at all levels of the source domain data. In the next step, there is a large amount of source domain data from the training model to transfer, and a small amount of target domain data for deep learning network fine-tuning can be used. In the third step, the transfer learning model is used for the target domain's and the source domain's common features. Here, we partly join the unique characteristics of the target domain with the common features obtained by the deep learning model, and then we input the data into the classifier for classification.

This paper will be divided into the following sections. Section 2 will introduce the deep learning framework (DBN) and the transfer learning method adopted in this paper. Section 3 will introduce the combination of the common features of the target domain and the source domain with the unique characteristics of the target domain method. Section 4 introduces the main steps of the experimental simulation's details, and Section 5 illustrates the comparative analysis of the experimental results.

## 2. Algorithm

The specific flowchart is shown in Figure 1. Our method is divided into two parts. In the first part, we use the deep transfer model to extract and transfer spectral features. In the second part, we combine the spatial features and spectral features to classify the data. In the first part, we first combine the deep belief network and transfer learning theory to extract and transfer hyperspectral

data spectral features. In this part, we use source domain data that contains a large amount of data and labels to train the network. We can get the network model's structural parameters, network weights, and offsets after network training; then, we use the transfer learning theory to transfer the network model parameters. The deep belief network model can gradually extract data—from the specific features to the abstract features—from low to high levels. The lower layer of the source domain data model is the transfer layer; the network structure and the corresponding network parameters are transferred to the target domain data model, and the target domain high-level parameters are randomly initialized. The network output layer is connected to the logic regression layer, and finally, the small target domain label data is used to adjust the network parameters. We extract the last layer of the hidden layer output as the spectral feature of the data. In the second part, principal component analysis (PCA) is used to reduce the dimension of the target data and extract the main features of the data. We cut the data after dimensionality reduction into a data cube as the spatial feature of the data; then, we expand the data cube into a one-dimensional vector. The spectral information obtained in the previous part is combined with the spatial features into the fused feature information. The fused features are used as the input of the classifier to realize the recognition and classification of the object.

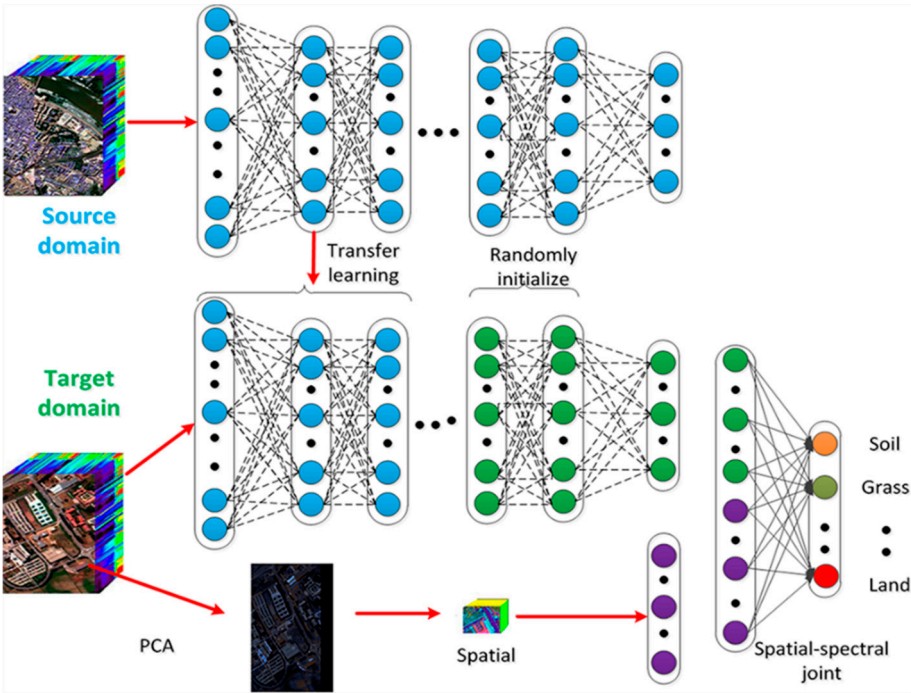

**Figure 1.** Transfer learning algorithm diagram.

We use the source domain data for the deep learning network training, and we use the target domain's limited labeled data for fine-tuning the network. The deep learning network is used for the process of extracting features from a low to high level in the corresponding feature extraction. Then, ultimately, we answer the question regarding the effect of the corresponding transfer learning. In this paper, we simulate the classification of the above data for the final target domain for pre-training the network structure.

### 2.1. Deep Belief Network

A deep belief network is an unsupervised deep learning network [23]. There are multiple layers of restricted Boltzmann machines stacked together. A deep learning network is composed of a multi-layer neural network overlay. Through the vertically stacked multi-layer network structure, the layer and neurons are connected to each other and correspond to the weights and the paranoid values

as structural parameters. The restricted Boltzmann machine is a special topology of the Boltzmann machine (BM). The topology of the Boltzmann machine network is shown in Figure 2 [24].

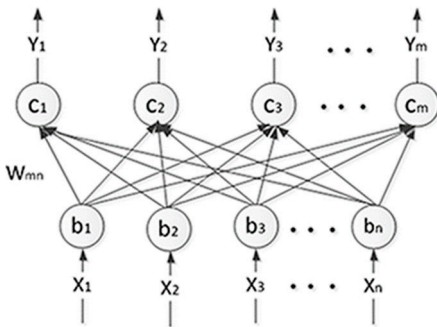

**Figure 2.** Restricted Boltzmann machine network topology.

The restricted Boltzmann machine network consists of two layers, namely, the visual and hidden layers. The neurons between the two layers are fully connected, and the neurons in the same layer are independent of each other. Meanwhile, both satisfy the Boltzmann distribution, which can model any discrete distribution [20] (see Figure 3). The conditional probability approach is used, which makes the two layers have a mutual reconstruction. By adjusting the parameters, when the difference between the hidden reconstruction vector and the original input sample is less than a set minimum value, the lowest energy function is found to be the end of training. Furthermore, the network training is performed to obtain the three types of parameters, namely, the network weights $W_{mn}$ and the offsets $B_n$ and $C_m$, as follows:

$$B_n = \begin{bmatrix} b_1 & b_2 & \dots b_n \end{bmatrix} \tag{1}$$

$$C_m = \begin{bmatrix} c_1 & c_2 & \dots & c_m \end{bmatrix} \tag{2}$$

The energy equation is:

$$E(x,y,\theta) = -\sum_{ij} w_{ij} x_i y_j - \sum_i b_i x_i - \sum_j c_j y_j \tag{3}$$

where $\theta = \{w, b, c\}$.

The joint probability density is:

$$p(x,y) = \frac{e^{-E(x,y)}}{\sum\limits_{x,y} e^{-E(x,y)}} \tag{4}$$

Since the probability is a special Gibbs probability distribution, according to the relevant mathematical knowledge, it can be obtained as follows [25]:

$$\frac{\partial \ln p(x)}{\partial w_{ij}} = x_j p(y_i = 1|x) - \frac{1}{l}\sum_k^l p(y_i = 1|y_k')y_{kj}' \tag{5}$$

$$\frac{\partial \ln p(x)}{\partial b_j} = x_j - \frac{1}{l}\sum_k^l y_{kj}' \tag{6}$$

$$\frac{\partial \ln p(x)}{\partial c_i} = p(y_i = 1|x) - \frac{1}{l}\sum_k^l p(y_i = 1|y_k') \tag{7}$$

Thus, the weight update formula is as follows (where the learning rate $k$ is used in the first $k$ cycles):

$$w_{ij} = w_{ij} + \alpha \left( x_j^{(0)} p\left( y_i = 1 \middle| x^{(0)} \right) - x_j^{(k)} p\left( y_i = 1 \middle| x^{(k)} \right) \right) \tag{8}$$

$$b_j = b_j + \alpha \left( x_j^{(0)} - x_j^{(k)} \right) \tag{9}$$

$$c_i = c_i + \alpha \left( p\left( y_i = 1 \middle| x^{(0)} \right) - p\left( y_i = 1 \middle| x^{(k)} \right) \right) \tag{10}$$

Where:

$$p(y_i = 1|x) = sigmoid \left( \sum_{j=1}^{m} w_{ij} x_j + c_i \right) \tag{11}$$

The loss function is as follows:

$$J(\theta) = -\frac{1}{m} \sum_{i=1}^{m} \left[ y^{(i)} \log h_\theta \left( x^{(i)} \right) + \left( 1 - y^{(i)} \right) \log \left( 1 - h_\theta \left( x^{(i)} \right) \right) \right] \tag{12}$$

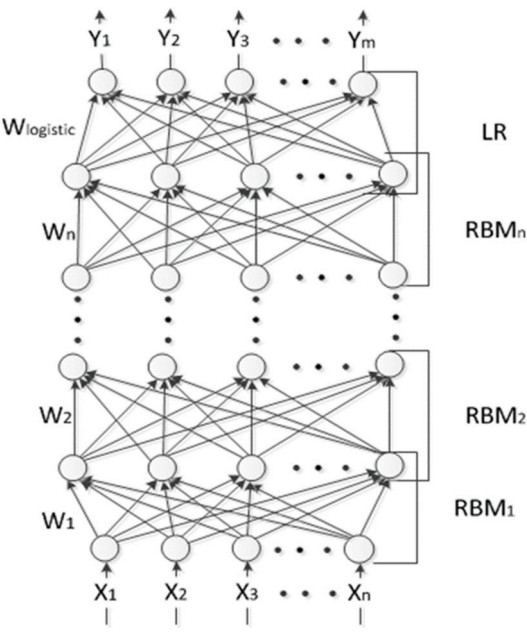

**Figure 3.** Deep belief network topology.

The initialization of the parameters of the deep learning network can be seen as the sequential training of several limited two-layer restricted Boltzmann machines from the input layer to the last hidden layer. The network formed by this training method is called a deep belief network. The purpose of this method is to overcome the problem of gradient diffusion. In deep belief network topology, which is shown in Figure 3, the top-level logistic regression network parameters are randomly generated, and do not require training [26,27]. The deep belief network works from the bottom to the top of the greedy unsupervised learning. The aim is to fit all of the features of the data as much as possible. In addition, the clustering integration accuracy, robustness, and stability increase, and the data can be divided.

*2.2. Transfer Learning*

Transfer learning is a new idea proposed in machine learning. Transfer learning uses existing knowledge that has been learned to transfer the original learning experience to a new learning model when encountering similar problems [28]. There are many examples of transfer learning that people

use in their learning of various types of knowledge in nature. For example, when one learns to ride a motorcycle, it is easier to learn first by riding a bicycle. It is also a great advantage when one learns to play the guitar after learning to play the piano.

At present, the research on transfer learning is mainly divided into several aspects. First, sample transfer generally means weighting the samples and giving more weight to the more important samples [29]. Second, feature transfer generally needs to project the features of the source and target areas into the same feature space [30]. The feature transfer is based on observing the common features between the images of the source domain and the target domain, and then using the observed common features to automatically transfer between different levels of features [31]. Third, when the whole method of model transfer is applied to the target area, fine-tuning of the pre-trained deep network is commonly used. This process can also be called parameter transfer.

For hyperspectral data, the deep learning model can achieve even greater accuracy regarding feature expression and classification. However, deep learning has some problems, such as its long training time and the large amount of sample data required for training. In this paper, the model-based transfer learning method, which is the combination of transfer learning and deep learning, can greatly save the deep learning model training time, and it also reduces the number of required labeled samples, which presents significant savings of manpower expenditures, materials, and financial resources during the annotation of the hyperspectral data samples.

We propose a DBN-based transfer learning network model. The purpose is to use the DBN network structure to obtain the common characteristics of the source domain data and the target domain data (see Figure 4).

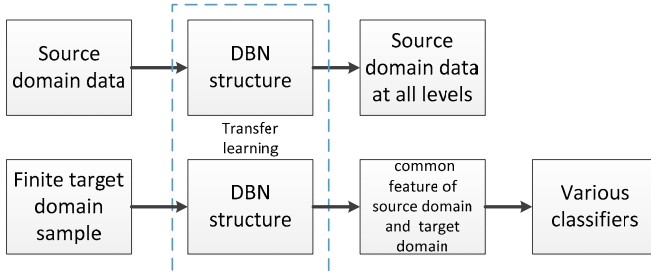

**Figure 4.** Deep belief network (DBN) transfer structure.

The schematic diagram of the network structure is shown in Figure 5. First, the source domain data are obtained. The source domain data are characterized by a sufficiently large amount of data to train the network parameters, and can be used to fine-tune the network structure (see Figure 5). The source domain data are used to construct a limited Boltzmann machine DBN network model to determine the hidden layer network number and the number of neuron nodes. The network's weight parameters and bias parameters are iterated through the labeled data. Then, the first few layers that have been trained by the DBN network are extracted as the target domain network. The weights of the high-level data network in the target domain are fine-tuned by means of random initialization (see Figure 6).

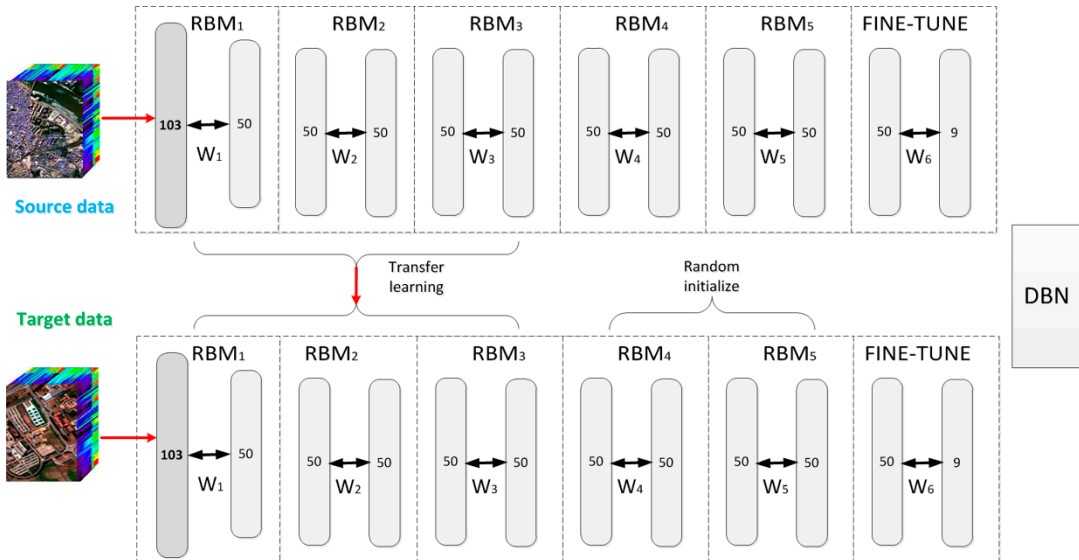

**Figure 5.** DBN spectral feature transfer method.

The corresponding model is transferred based on the deep learning network. The entire model is transferred from the lowest layer to the top layer to the target domain. Different layers have different transfer ability. The transferability of different levels has an important impact on the results of transfer learning, and thus needs to be explored. In the flowchart in Figure 6, firstly, a deep learning network model is constructed by using the source domain data set, and the structure and model parameters of the network model are transferred. The number of layers to be transferred is gradually increased from small to large. According to the accuracy of the final classification, the number of layers to be transferred by the transfer learning model is determined.

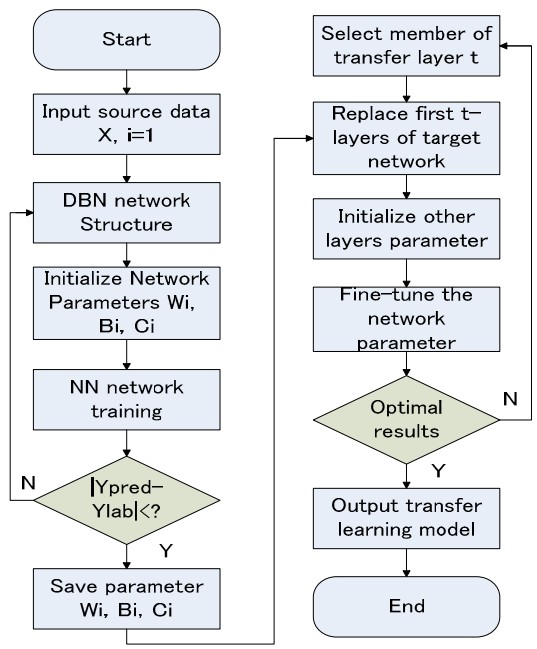

**Figure 6.** DBN spectral feature transfer algorithm flowchart.

Different samples and different spectral frequencies all reflect different features. For a large number of pre-trained hyperspectral data, the deep learning network is composed of a multi-layer neural network that corresponds to a number of network structural parameters. These network

structural parameters reflect low to high abstractions with different categories of features. The deep learning network is composed of several limited Boltzmann machines with components that extract a plurality of hyperspectral data characteristics. The transfer learning algorithm is as follows:

---

**Algorithm 1.** deep transfer learning training

---

1: **Initialize** $\{W_{si}, B_{si} \text{ and } C_{si}\}$, $\theta = W, B, C, \text{ for } i = 1, 2, 3, 4, \ldots, n$;
2: **Input** X
3: **for** i = 1,2, ... , n;
4: 　　Train $i$ *RBM*
5: 　　$x_{i+1} = sigmoid(c_i + x * w_i)$
6: **end**
7: **If** Fine tune training
8: 　**while** $ij \, \Delta \varepsilon \leq$ *Threshold*
9: 　　　Input $\{\mathbf{X'}; \mathbf{Y}\}$
10. 　　　Initialize $W_i (0 < W_i < 1)$
11: 　　　Using Gradient Descent method
12: 　　　Output (k)(k) $\{Ws, bs\}, (k = 1, \ldots, n)$
13: 　**end**
14: **End**
15: **INPUT** transfer layers T
16: **FOR** T = 1,2,3, ... , n;
17: 　　　**Replace** parameters $\{W_{si}, B_{si}\} \rightarrow \{W_{Ti}, B_{Ti}\} \, for \, i = 1, 2, 3, 4, \ldots, T$;
18: 　　　**Initialize** $W_{Ti}, B_{Ti} \text{ and } C_{Ti} \, for \, i = T + 1, \ldots, n$;
19: 　　　Fine-tune network using gradient descent method;
20: 　　　Output optimal results, parameter T, Output (k)(k) $\{W_T, B_T\}, (k = 1, \ldots, n)$
21: **END**

---

## 3. Experiment and Parameter Selection

### 3.1. Experimental Data

The source data used in this experiment are from the Pavia Metropolis hyperspectral dataset (http://www.ehu.eus/ccwintco/index.php?title=Hyperspectral_Remote_Sensing_Scenes&oldid=16576). The Pavia Metropolitan Hyperspectral Data come from the University of Pavia Telecommunications and Remote Sensing Laboratory in Italy, and were provided by Prof. Paolo Gamba. Their image consists of a 1096 × 1096 pixel composition, the spectral resolution (band number) is 102, the ground object resolution is approximately 1.3 m, and the processed data contain nine different categories. The black portion of the figure was removed as invalid data. The spectral cube data were taken by ROSIS (a German aerial spectrograph) in Pavia, northern Italy (see Figure 7).

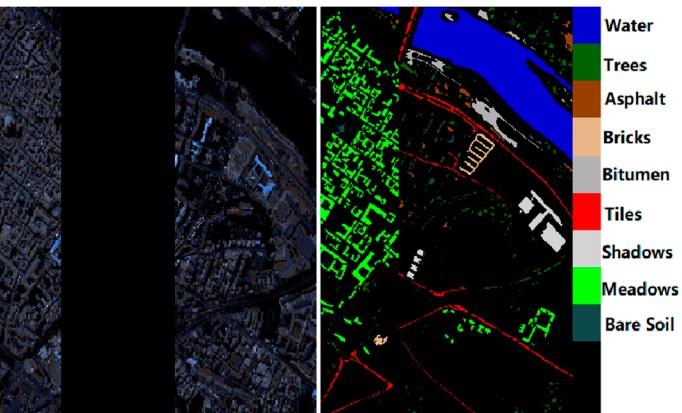

**Figure 7.** Pavia center false color image (left) and ground truth image (right).

The target dataset uses the hyperspectral data cube of the University of Pavia that was photographed by ROSIS (a German aerial photodetector) in the city of Pavia, northern Italy, which is the measured data (http://www.ehu.es/ccwintco/index.php?title=Hyperspectral_Remote_Sensing_Scenes&oldid=16576). In the image, Pavia University consists of $610 \times 610$ pixels with a spectral resolution (band number) of 103 and a ground object resolution of approximately 1.3 m. The processed data contain nine different categories for training and classification (see Figure 8). The pure black part was removed as invalid data (see Figure 9).

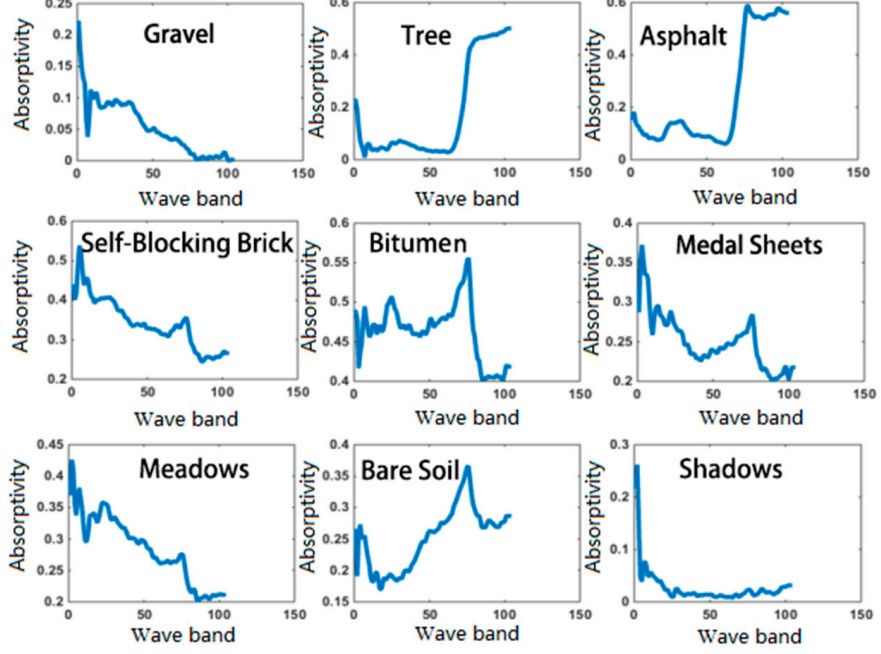

**Figure 8.** Nine kinds of spectral curves of ground object categories.

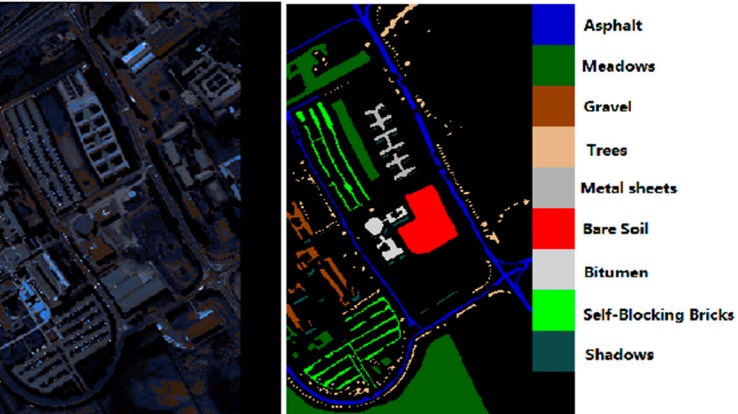

**Figure 9.** Pavia University false color image (left) and ground truth image (right).

## 3.2. Influence of Network Parameter

This part of the experiment uses the source domain data for experimental verification through a grid search for the number of hidden layers and the number of neurons per hidden layer. The influence of the depth of the neural network and number of neural nodes of DBN [32] on the research results is analyzed (see Figure 10). It can be seen from Figure 10 that when the other factors are unchanged, the highest accuracy is obtained when the number of deep neural networks is five. This means that the network structure contains five Restrict Boltzmann Machine (RBM)s. Thus, when the number of network layers is too small (especially when the number of layers is less than five), the complexity of the structure of the deep neural network is too low to extract the high-level features in the data, and the acquired features are poorly represented. This will result in a large deviation, because there has been a lack-of-fit phenomenon. When the number of layers is too large, the classification effect also deteriorates. This is because as the number of network structural layers increases, the structure becomes increasingly more complicated, and the over-fitting phenomenon occurs. Therefore, it is necessary to determine the number of layers in the neural network according to the specific size of the data in the process of constructing the structure of the deep neural network. If the number of layers is too high or too low, the classification result is unfavorable.

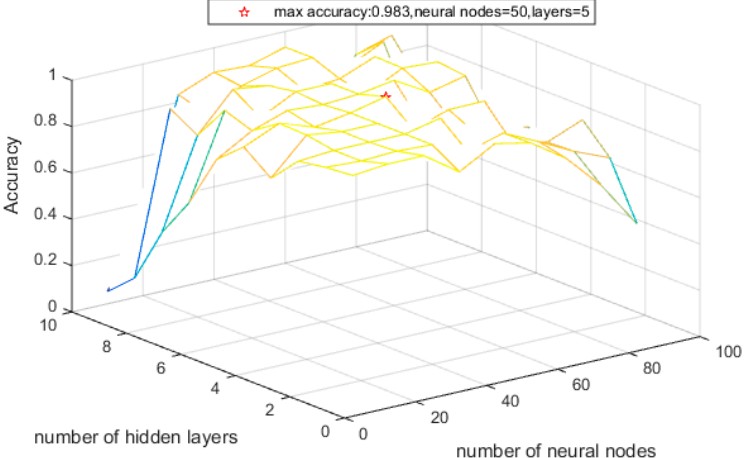

**Figure 10.** Pavia center data classification accuracy grid search with the number of hidden layers and number of neural nodes.

The width of the deep neural network has an important influence on the experimental results. The main methods used in the current research are to determine the corresponding network width using a continuous test. The highest accuracy is a five-layer network structure and 50 neural nodes.

As seen from the figure, when the DBN network width is 50 neuron nodes, the optimal classification results occur at 98.3%. When the number of network neuron nodes is smaller than the optimal number of nodes, the complexity of the network is low, and the number of parameters is small. Although the training time and the accuracy of the network are low, the simple model means that the extracted features are simple and low, and the low accuracy of the network at this time is due to the network deviation. However, when the number of neuron nodes is higher than the optimal number of nodes, the model's language is complicated, and the data classification accuracy of the test set is not high enough for the training data to be well written. The model's generalization ability is poor, and the model suffers from the over-fitting phenomenon.

### 3.3. Weight Visualization

Moreover, we can visualize the connection between the input layer and the hidden layer in order to improve the depth of the network to study the features of the hyperspectral data. There are no connections between the input neurons and the neurons of the hidden layer, and there are no connections between the neurons of the hidden layer. Assuming that the N-dimensional spectral curve vector of the input layer and the number of neurons in the hidden layer are M, then each neuron of the hidden layer is connected with the N neurons of the input layer. Each connection corresponds to a weight component, and each neuron has 1 * N weight and is transformed into a gray matrix. Then, there will be an N-dimensional gray matrix. The gray matrix can be seen as a feature extraction filter, which shows the depth of the learning network. The structure of the hidden layer has 50 layers, and then there are 50 filters. Since the input sample is 103-dimensional data, it will be converted to a 13 * 8 size filter, as shown on Figure 11.

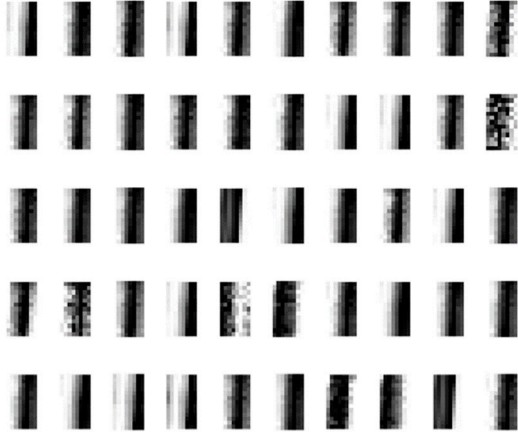

**Figure 11.** DBN network weight visualization.

### 3.4. Influence of Transfer Layer

The number of transfer layers refers to the number of layers in the source domain data training model that is transferred to the target domain model. The source domain data and the training model can obtain low-level and high-level hyperspectral images and from simple to complex multilevel features. The common features between the two data sets are transferred to simplify the complexity of the target domain data network while addressing the lack of precision due to the insufficient sample size. In this paper, the simulation experiments determine the highly compatible features of the characteristics, the transfer ability, the layers of poor generalization ability [33], and what is not conducive to transfer. The simulation results regarding accuracy changes with the number of transfer layers are shown in Table 1 and Figure 12, with abbreviation of Transfer (TSF) and Logistic Regression (LR).

**Table 1.** Deep neural network classification of different transfer results.

| Method | The Number of Layers from the Source Domain Data to the Target Domain Data | | | | | |
|---|---|---|---|---|---|---|
| | **0** | **1** | **2** | **3** | **4** | **5** |
| TSF-DBN-LR | 0.891 | 0.9012 | 0.9283 | 0.9108 | 0.8831 | 0.8666 |

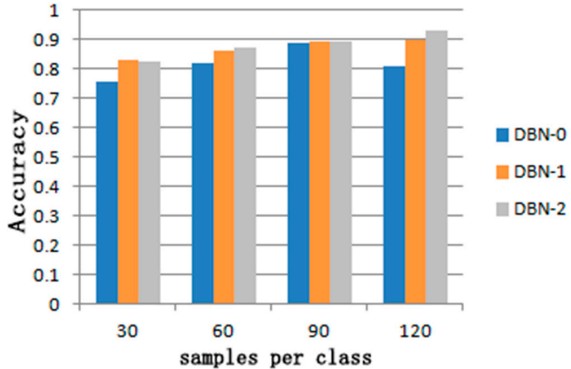

**Figure 12.** Accuracy changes with the number of transfer layers.

### 3.5. The Number of Target Domain Samples

The sample size of the target domain data set has an important impact on transfer learning. The size of the sample directly affects the generalization ability of the model. The simulation results of different scale samples give the following results:

Table 2 shows the classification results of the two transfer learning methods as a function of the sample size. The M in DBN-M represents the number of transfer layers. M = 0 indicates that the results of the transfer learning method are not used. Figure 13 shows the results for each sample size of 30, 60, 90, and 120. From the trend of the sample size in the figure, the accuracy of the model varies with the sample size. When the sample size is 30 and 60 per class, the transfer learning has a good effect, and the effect of improving the accuracy is obvious. With the increase in the sample size, when the number of samples reaches 90 and 120, the effect of transfer learning on the model's results is not obvious.

**Table 2.** The classification results of different target regions' transfer learning based on the sample size.

| Sample Size | DBN Transfer Model | | |
|---|---|---|---|
| | **DBN-0** | **DBN-1** | **DBN-2** |
| 30 | 0.759 | **0.832** | 0.824 |
| 60 | 0.821 | 0.862 | **0.872** |
| 90 | 0.886 | **0.895** | 0.893 |
| 120 | 0.81 | 0.9012 | **0.9283** |

**Figure 13.** Change in accuracy versus target domain sample.

## 4. Experimental Results Comparation

### 4.1. Spatial and Spectral Information Fusion

Hyperspectral data are three-dimensional image data that include the spectral information of pixels and the spatial information of pixels, which is the spatial positional relationship between each pixel and the peripheral pixels [34,35]. In the process of hyperspectral image processing, the spectral information and the spatial information have important influences on the final classification results. In this paper, first the deep neural network is used to classify the spectral information of the hyperspectral data. Then, the PCA algorithm is used to extract the data using the reduced-dimension spatial neighborhood of the data features. The simulation results of the accuracy changes with the sample proportion changes of the spectral feature, spatial feature, and joint spectral and spatial features (joint spe–spa) are shown in Table 3 and Figure 14. Table 3 and Figure 14 show that the spatial information and spectral information are combined in order to improve classification accuracy.

**Table 3.** Comparison of the accuracy changes with the sample proportion changes of the spectral characteristics, spatial characteristics, and space-spectrum binding characteristics (joint spe–spa).

| Sample Proportion | Spectral Feature | Spatial Feature | Joint spe–spa |
|---|---|---|---|
| 10% | 0.8532 | 0.8743 | 0.8921 |
| 20% | 0.8823 | 0.8967 | 0.9108 |
| 30% | 0.9011 | 0.9101 | 0.9236 |
| 40% | 0.9243 | 0.9283 | 0.9421 |
| **50%** | **0.9291** | **0.9323** | **0.9460** |

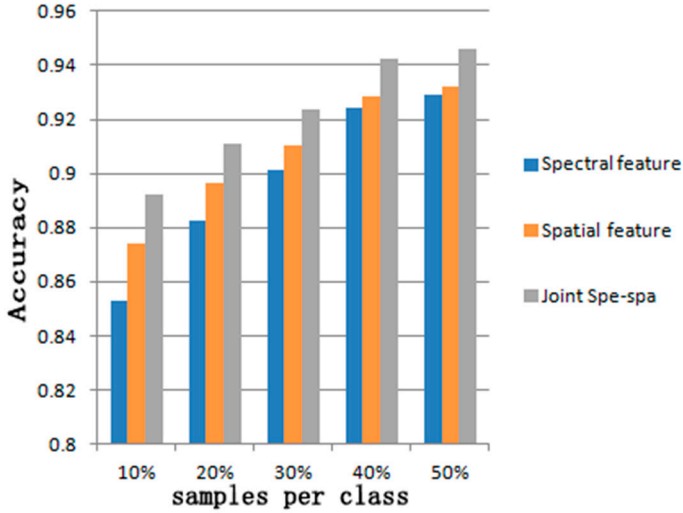

**Figure 14.** Space spectrum information fusion and accuracy contrast.

### 4.2. Comparison with Other Methods

The comparison of different classification algorithms and dimensionality reduction algorithms is shown on Table 4, and the comparison of different classification algorithms is shown in Figure 15. Table 4 shows a comparison between the deep neural network algorithm and the traditional classification algorithm. The analysis and verification of different algorithms were conducted for the classification and identification of hyperspectral data set recognition. The deep neural network algorithm was constructed by setting the output layer to the fully connected layer for multi-class classification. In fact, the combination of feature extraction and the logistic regression algorithm can be seen as a deep learning algorithm with feature extraction and a data dimension reduction. The methods combine the extracted features with other algorithms, such as the Support Vector Machine, Naive Bayesian Model, K Nearest Neighbor (SVM, NBM, KNN), and others.

As seen from Figure 15, when the data size is increased to 30 per class, by comparing the classification results of different algorithms, it can be seen from the results of the above figure that deep learning combined with the SVM algorithm can achieve better classification results by using transfer learning and the deep learning algorithm DBN. The deep transfer learning model can effectively improve the classification accuracy compared with the traditional feature extraction PCA algorithm. Compared with the data without the dimensionality reduction, the dimensionality reduction and feature extraction process can effectively reduce the complexity of the data and improve the data divisibility so that the classification accuracy is greatly improved. Therefore, it can be concluded that feature extraction can obviously improve the data divisibility. Furthermore, different classification algorithms have a certain impact on the classification accuracy. In contrast, the SVM is the best SVM algorithm in comparison with the KNN and NBM algorithms.

The SVM classification results, the PCA-SVM classification results, the DBN-SVM classification results, and the DBN-SVM spatial classification results are shown in Figure 16. Through the comparison experiment, we can see that the proposed algorithm based on the transfer learning feature can achieve higher accuracy, and the accuracy can be improved by combining the spatial information, as informed in Section 3.1.

**Table 4.** Comparison of different classification algorithms and dimensionality reduction algorithms. PCA: principal component analysis.

| Method | Target Domain Dataset Label Sample Size (A/C) | | | |
|---|---|---|---|---|
| | **30** | **60** | **90** | **120** |
| SVM | 0.701 | 0.759 | 0.786 | 0.801 |
| NBM | 0.621 | 0.651 | 0.681 | 0.711 |
| KNN | 0.675 | 0.732 | 0.743 | 0.776 |
| PCA+SVM | 0.751 | 0.821 | 0.853 | 0.887 |
| PCA+NBM | 0.701 | 0.753 | 0.761 | 0.811 |
| PCA+KNN | 0.731 | 0.802 | 0.831 | 0.867 |
| **DBN+SVM** | **0.832** | **0.862** | **0.895** | **0.9012** |
| DBN+NBM | 0.762 | 0.843 | 0.825 | 0.847 |
| DBN+KNN | 0.783 | 0.832 | 0.879 | 0.889 |

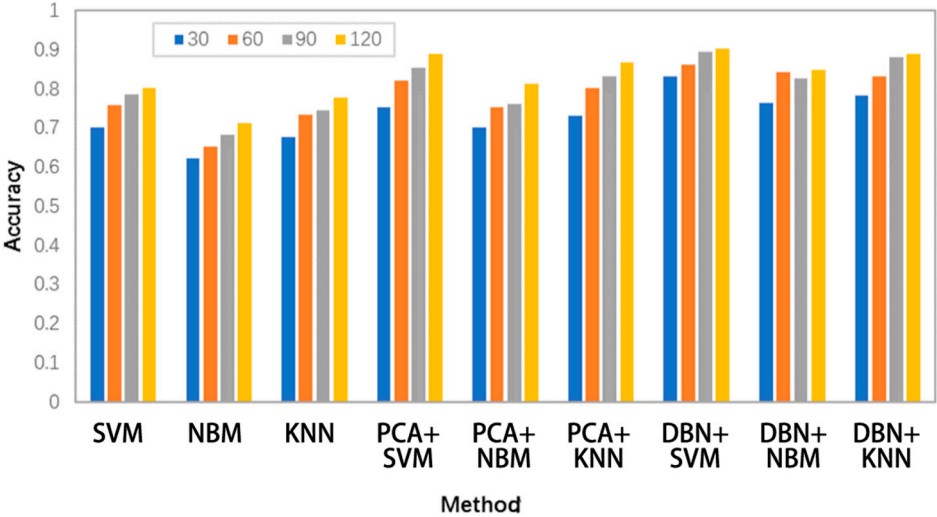

**Figure 15.** Comparison of different classification algorithm results.

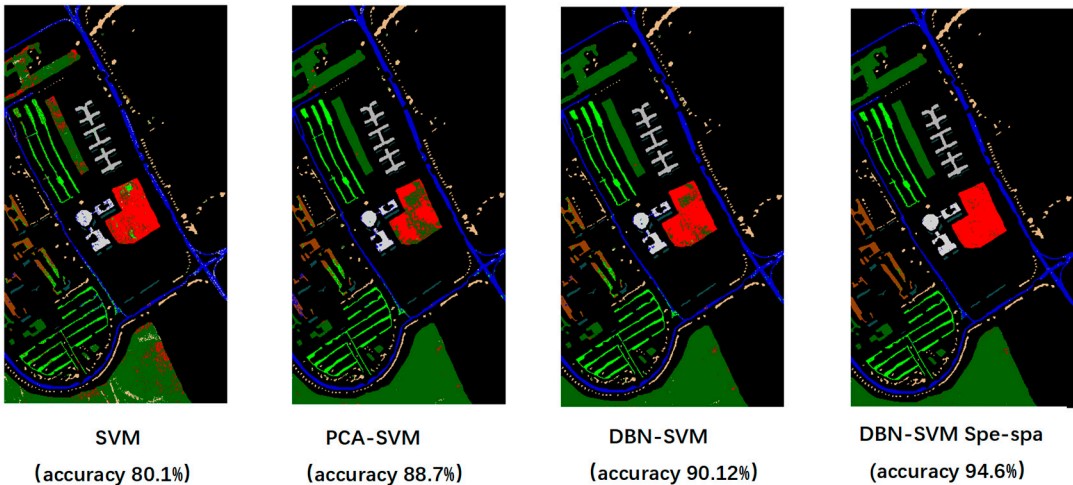

SVM
(accuracy 80.1%)

PCA-SVM
(accuracy 88.7%)

DBN-SVM
(accuracy 90.12%)

DBN-SVM Spe-spa
(accuracy 94.6%)

**Figure 16.** The above four pictures are the SVM classification results, the PCA-SVM classification results, the DBN-SVM classification results, and the DBN-SVM spatial classification results.

*4.3. Confusion Matrix*

To use a more intuitive statistical multi-classification effect, this paper uses the confusion matrix to describe the classification results of each category. Each column of the confusion matrix represents the prediction category, and the total number of each column indicates the number of data predicted for this category. Each row represents the true category of the data, and the total number of data for each row indicates the number of data instances for that category. The confusion matrix of without transfer, after transfer, and joint spe–spa is shown in Figure 17

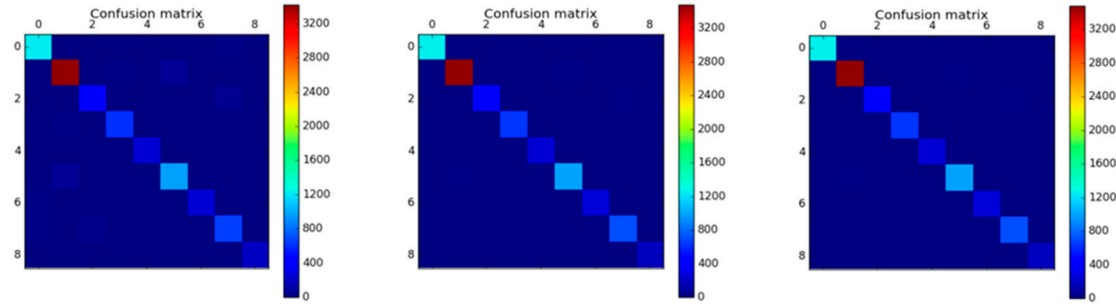

**Figure 17.** Confusion matrix. Without transfer (left), after transfer (mid), joint spe–spa (right).

As shown in Figure 17, the color of the block is most prominent, thus indicating the validity of the classification results during the experimental verification.

## 5. Conclusions

In this paper, a new method based on the combination of transfer learning and deep learning to acquire the common features and unique features is proposed. First, we obtain a large number of DBNs trained by source domain data. In this paper, according to the hierarchical features of the DBN networks, we use a limited sample tag to fine-tune the network. This experiment studies the effect of the features at all levels on the final classification results. Finally, the features extracted from the first few layers of the DBN network are low-level features, and the source domain data and the target domain data have good transfer. As a result, the more complex the extracted features, the less effective the transfer learning. Based on the simulation experiment, this paper concludes that transfer learning is optimal when the number of hidden layers is five.

Hyperspectral data are tested for simulations to find a model that performs better with fewer samples. Transfer learning can be used to obtain the common characteristics of the target domain and the source domain data, but each data set will inevitably have its own unique characteristics. Using only transfer learning will lose this part of the characteristics. For this purpose, the main components extracted by the PCA algorithm are mostly characteristic of the target domain data. After the experimental simulation, it can be concluded that the more main component layers that are used, the higher the classification accuracy will be. However, after the main component layers reach six layers, the accuracy rate increases more slowly. Therefore, a six-layer principal component cell is used. For the algorithm structure we proposed in this paper, we use the LR and SVM to separately test it. It can be seen from the comparative experiments that the proposed structure in this paper has a good effect in both the LR and SVM algorithms. Our future work will focus on the use of more deep learning models to conduct the transfers, testing, and extraction of the characteristics of the deep learning model in the transfer learning model framework. The model performances will be assessed and compared to the other methods.

**Author Contributions:** Conceptualization: K.L. Data curation: W.L. Formal analysis: K.L., M.W., Y.L. Funding acquisition: K.L., W.L. Investigation: Y.L. Methodology: W.L., K.L. Project administration: W.L., K.L. Resources: K.L. Software: M.W., Y.L., N.Y. Supervision: K.L. Validation: K.L., W.L. Visualization: K.L., W.L. Writing—original draft: W.L., K.L. Writing—review & editing: K.L., W.L.

**Funding:** This research received no external funding.

**Acknowledgments:** The authors are supported by the Chinese National Natural Science Foundation (No. 61773039), the Aeronautical Science Foundation of China (No. 2018XXX), and 'Fanzhou' Scientific Funds (No. 2017XX).

**Conflicts of Interest:** The authors declare no conflict of interest.

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
