# Peer review of "A Novel Method of Hyperspectral Data Classification Based on Transfer Learning and Deep Belief Network"

_applsci, doi:10.3390/app9071379_

Round 1

Reviewer 1 Report

This paper covers using transfer learning with DBNs for hyperspectral image classification.  I'm unsure of how novel it is because the literature base is so large, but a thorough double-check of the literature for existing work should be performed by the authors before publication.  The authors show how training DBNs on one dataset can be used with transfer learning to improve classification accuracy.  

As someone who's not a hyperspectral data expert, I don't understand why you can't just downsample the resolution and avoid the 'dimensionality disaster'.  I'm well aware of the curse of dimensionality, but for a dataset like this with a spatial resolution component, I would think you could just downsample the resolution to reduce dimensions.  I think this should be addressed in your intro.

I also think in your abstract/intro you could talk about more of the direct application in the paper -- classification of types of surfaces on the Earth.

When you talk about the sections in the intro, you start with the second section.  Is the first section assumed to be the introduction?  It isn't clear.

In your intro you said PCA doesn't work well due to the nonlinearity, but then you use it in your method.  It's not clear why this is added to the last layer either.  Is this based on existing work or something novel you came up with?  Does it actually help performance and what are the reasons for doing this?

It's better to use the sigma character than the word 'sigmoid' in equation 11.

A reference for the dataset in-line in the text would be a good idea.

You have some typos in there (missing spaces); be sure to spell check. For example "Thesenetwork" on page 8.

Fig 7 has a black band in the left image -- seems like an error.  The 'meadows' label also makes no sense -- that looks like buildings in the image.  Double-check your labels in the image.

I guess Fig 10 looks kind of neat as 3D, but a 2D heatmap would be cleaner and easier to interpret as well as see all info from the figure.  It's also missing a colorbar.  If color is just showing the z-axis, it's not necessary unless you want to be flashy.

Fig 15 should have all text shown for the x-labels and not have any cut off.

Fig 16 seems unnecessary -- all images look almost the same.  Maybe keep as supplementary info.

Author Response

This paper covers using transfer learning with DBNs for hyperspectral image classification.  I'm unsure of how novel it is because the literature base is so large, but a thorough double-check of the literature for existing work should be performed by the authors before publication.  The authors show how training DBNs on one dataset can be used with transfer learning to improve classification accuracy.  

As someone who's not a hyperspectral data expert, I don't understand why you can't just downsample the resolution and avoid the 'dimensionality disaster'.  I'm well aware of the curse of dimensionality, but for a dataset like this with a spatial resolution component, I would think you could just downsample the resolution to reduce dimensions.  I think this should be addressed in your intro.

Ans: downsample will cause spatial detail lost rapidly.

I also think in your abstract/intro you could talk about more of the direct application in the paper -- classification of types of surfaces on the Earth.

Ans: The original data has been classified already.

When you talk about the sections in the intro, you start with the second section.  Is the first section assumed to be the introduction?  It isn't clear.

Ans: It shouldn’t be introduction, we have fixed.

In your intro you said PCA doesn't work well due to the nonlinearity, but then you use it in your method.  It's not clear why this is added to the last layer either.  Is this based on existing work or something novel you came up with?  Does it actually help performance and what are the reasons for doing this?

Ans: PCA is used as a comparation, and it is also true that it is based on existing work, and the conclusion is PCA can’t achieve good effect due to the nonlinearity.

It's better to use the sigma character than the word 'sigmoid' in equation 11.

Ans: It’s not a sigma, it is a function.

A reference for the dataset in-line in the text would be a good idea.

Ans: We have added.

You have some typos in there (missing spaces); be sure to spell check. For example "Thesenetwork" on page 8.

Ans: Apologize for spell wrong.

Fig 7 has a black band in the left image -- seems like an error.  The 'meadows' label also makes no sense -- that looks like buildings in the image.  Double-check your labels in the image.

Ans: Original data have been deleted some information by Prof. Paolo Gamba.

I guess Fig 10 looks kind of neat as 3D, but a 2D heatmap would be cleaner and easier to interpret as well as see all info from the figure.  It's also missing a colorbar.  If color is just showing the z-axis, it's not necessary unless you want to be flashy.

Ans: We thought 3D will show all details, 2D will lose some information.

Fig 15 should have all text shown for the x-labels and not have any cut off.

Ans: We have redrawn the X-labels.

Fig 16 seems unnecessary -- all images look almost the same.  Maybe keep as supplementary info.

Ans: The left indeed have some differences, maybe the middle and the right look too same but we don’t want to lose information.

Reviewer 2 Report

This paper describes and evaluates a hyperspectral classification technique that makes use of deep belief networks and transfer learning. The overall technique is explained in this paper and the evaluation has been done with a single dataset.

While this paper seems to be promising, there are some concerns that need to be addressed. The comments are provided below.

1)     It would be good to provide a name for this novel technique since the authors claim that the combination of DBN and transfer learning is new for this domain.

2)     The authors incorrectly describe Figure 1 as a flowchart but the diagram is just a representation of the transfer learning algorithm

3)     The Pavia Metropolitan Hyperspectral dataset is used as the source data. The authors need to provide a citation and link for this dataset. Also, the authors need to give a justification on why this dataset is used. What about other datasets?

4)     The target dataset is also from the University of Pavia. A link and citation could also be provided for this as well.

5)     Is the Figures 8 accurately described? The category names seem to be missing

6)     In line 253, it is written that the optimal classification accuracy reached was 98.3%. However, in the Figure 10, the max accuracy reached seems to be around 90%. Please check this discrepancy.

7)     The results in Table 1 are not sufficiently discussed.

8)     The results in Table 4 and Figure 15 are not sufficiently discussed. The authors just say the DBN+SVM provide the best results. The reason why this combination provided the best results need to be discussed.

9)     Citations and full algorithm names need to be provided in Line 235

10)  The limitations are not highlighted in the Conclusions section. They need to be added.

Other minor corrections

a)      Transfer spelling wrong at the end of Line 87

b)     In Line 175, it is mentioned transfer learning model-based transfer learning method. This needs to be corrected

c)      In Line 207, add space between ‘these’ and ‘network’

d)     The introduction section starts with 0 at Line 29. This is not a standard practise. It should 1.

e)     In Line 295, ‘the number of layers transfer’ should be ‘number of transfer layers’?

f)       In Line 351, it is mentioned ‘multi-classification classification effect’. Delete the repeated word.

Author Response

While this paper seems to be promising, there are some concerns that need to be addressed. The comments are provided below.

1)    It would be good to provide a name for this novel technique since the authors claim that the combination of DBN and transfer learning is new for this domain.

Ans:we have name it in Line 26.

2)    The authors incorrectly describe Figure 1 as a flowchart but the diagram is just a representation of the transfer learning algorithm

Ans: We have changed the name.

3)    The Pavia Metropolitan Hyperspectral dataset is used as the source data. The authors need to provide a citation and link for this dataset. Also, the authors need to give a justification on why this dataset is used. What about other datasets?

Ans: We have given the link in Line 215 and 216.

4)    The target dataset is also from the University of Pavia. A link and citation could also be provided for this as well.

Ans: We have given the link in Line 215 and 216.

5)    Is the Figures 8 accurately described? The category names seem to be missing

Ans: We have added the category names.

6)    In line 253, it is written that the optimal classification accuracy reached was 98.3%. However, in the Figure 10, the max accuracy reached seems to be around 90%. Please check this discrepancy.

Ans: The figure has been redrawn.

7)    The results in Table 1 are not sufficiently discussed.

Ans; We have checked the table change the form so that we can see the LR accuracy.

8)    The results in Table 4 and Figure 15 are not sufficiently discussed. The authors just say the DBN+SVM provide the best results. The reason why this combination provided the best results need to be discussed.

Ans: We have given details, in fact it’s DBN+SVM spatial .

9)    Citations and full algorithm names need to be provided in Line 235

Ans: We have provided in Line 237.

10)  The limitations are not highlighted in the Conclusions section. They need to be added.

Ans: We have highlighted in Line 383 to 385.

Other minor corrections

a)     Transfer spelling wrong at the end of Line 87

Ans: We have fixed.

b)     In Line 175, it is mentioned transfer learning model-based transfer learning method. This needs to be corrected

Ans: We have corrected

c)     In Line 207, add space between ‘these’ and ‘network’

Ans: space added.

d)     The introduction section starts with 0 at Line 29. This is not a standard practise. It should 1.

Ans: We have fixed, now it’s 1.

e)     In Line 295, ‘the number of layers transfer’ should be ‘number of transfer layers’?

Ans: We have corrected.

f)       In Line 351, it is mentioned ‘multi-classification classification effect’. Delete the repeated word.

Ans: We have deleted.

Round 2

Reviewer 2 Report

Looks like the authors have rushed through the changes in the revised manuscript. For instance, a name for the model has been added only to the abstract and nowhere else in the document! The authors are expected to add this name in the relevant sections of the manuscript.

Is TDBNN a new model or algorithm? The wording is a bit confusing all through the document. Please try to be consistent with the wording.

The link to the source dataset has been added to the manuscript in Line 218. What about the link to target dataset? Does the same source also contain the target dataset? If so, the authors need to mention it in the manuscript.

The categories in the Figures 8 and Figure 9 do not match. Please address this discrepancy.

The below comments haven’t been properly addressed from my earlier review

1)      The results in Table 4 and Figure 15 are not sufficiently discussed. The authors just say the DBN+SVM provide the best results. The reason why this combination provided the best results need to be discussed.

2)      Citations and full algorithm names need to be provided in Line 235

Author Response

Dear Reviewer,

We delete TDBNN.

1)      The results in Table 4 and Figure 15 are not sufficiently discussed. The authors just say the DBN+SVM provide the best results. The reason why this combination provided the best results need to be discussed.

Ans: we discussed them on Page 13

2)      Citations and full algorithm names need to be provided in Line 235

Ans: we provided it.

Thank you so much!

Li 
